# The Identification of *AMT* Family Genes and Their Expression, Function, and Regulation in *Chenopodium quinoa*

**DOI:** 10.3390/plants13243524

**Published:** 2024-12-17

**Authors:** Xiangxiang Wang, He Wu, Nazer Manzoor, Wenhua Dongcheng, Youbo Su, Zhengjie Liu, Chun Lin, Zichao Mao

**Affiliations:** 1College of Agronomy and Biotechnology, Yunnan Agricultural University (YNAU), Kunming 650201, China; wxx20231016@163.com (X.W.); wh19990709@163.com (H.W.); rananazermanzoor@gmail.com (N.M.); wenhuadc@ynau.edu.cn (W.D.); lzj1022@163.com (Z.L.); 2College of Resources and Environment, Yunnan Agricultural University, Kunming 650201, China; youbosu@ynau.edu.cn; 3Institute of Improvement and Utilization of Characteristic Resource Plants, Yunnan Agricultural University, Kunming 650201, China; 4The Laboratory for Crop Production and Intelligent Agriculture of Yunnan Province, Kunming 650201, China

**Keywords:** quinoa, ammonium transporter, nitrogen assimilation, transcriptional factor, gene regulatory networks

## Abstract

Quinoa (*Chenopodium quinoa*) is an Andean allotetraploid pseudocereal crop with higher protein content and balanced amino acid composition in the seeds. Ammonium (NH_4_^+^), a direct source of organic nitrogen assimilation, mainly transported by specific transmembrane ammonium transporters (*AMTs*), plays important roles in the development, yield, and quality of crops. Many *AMTs* and their functions have been identified in major crops; however, no systematic analyses of *AMTs* and their regulatory networks, which is important to increase the yield and protein accumulation in the seeds of quinoa, have been performed to date. In this study, the *CqAMTs* were identified, followed by the quantification of the gene expression, while the regulatory networks were predicted based on weighted gene co-expression network analysis (WGCNA), with the putative transcriptional factors (TFs) having binding sites on the promoters of *CqAMTs*, nitrate transporters (*CqNRTs*), and glutamine-synthases (*CqGSs*), as well as the putative TF expression being correlated with the phenotypes and activities of GSs, glutamate synthase (GOGAT), nitrite reductase (NiR), and nitrate reductase (NR) of quinoa roots. The results showed a total of 12 members of the *CqAMT* family with varying expressions in different organs and in the same organs at different developmental stages. Complementation expression analyses in the triple mep1/2/3 mutant of yeast showed that except for *CqAMT2.2b*, 11/12 *CqAMTs* restored the uptake of NH_4_^+^ in the host yeast. *CqAMT1.2a* was found to mainly locate on the cell membrane, while TFs (e.g., *CqNLPs*, *CqG2Ls*, *B3* TFs, *CqbHLHs*, *CqZFs*, *CqMYBs*, *CqNF-YA/YB/YC*, *CqNACs*, and *CqWRKY*) were predicted to be predominantly involved in the regulation, transportation, and assimilation of nitrogen. These results provide the functions of *CqAMTs* and their possible regulatory networks, which will lead to improved nitrogen use efficiency (NUE) in quinoa as well as other major crops.

## 1. Introduction

Nitrogen (N), with the major forms being nitrate (NO_3_^−^) and NH_4_^+^, is an important essential macronutrient for the growth and development of crops [1]. As sessile organisms, plants have evolved a variety of adaptive mechanisms to respond to internal and external nitrogen availability [2]. Research has showed that the concentration of NH_4_^+^ is much lower than that of NO_3_^−^ in aerobic soils; however, NH_4_^+^ richness was found in flooded and acidic soils, while NH_4_^+^ is a major source of nitrogen absorbed by plants such as paddy rice [3,4,5,6]. The *AMTs* are important protein carriers known to absorb and transport the NH_4_^+^ that maintains ammonium homeostasis among the organs and tissues of plants [7,8,9]. According to the Michaelis–Menten constant (km) of *AMTs* to NH_4_^+^, *AMTs* can be divided into high-affinity transporters (HATs), with Km < 100 µM, and low-affinity transporters (LATs), with a non-saturating kinetics propriety [8,10,11]. The HATs mediate the high-affinity absorption of NH_4_^+^ from soils with limited concentrations [9,10], playing a crucial role in NH_4_^+^ absorption, followed by its assimilation and redistribution, while the LATs can quickly and efficiently transport or export NH_4_^+^, especially when there is significant accumulation of the NH_4_^+^ within the cell or tissue, which may interfere with the normal primary metabolisms, such as the citric acid cycle and processing of oxidative phosphorylation for energy production [12]. The identified *AMTs* have different locations and various expression levels among the organs as well as in the same organ at different developmental stages. The *AMTs* may be particularly important for organogenesis and development regulation through regulation of the uptake, assimilation, and organic redistribution of the nitrogen source, in which TFs and nitrogen signaling are involved in the regulation of *AMT* expression and function [13,14,15]. The first plant NH_4_^+^ transporter gene (*AtAMT*) was identified in *Arabidopsis thaliana* by the restored NH_4_^+^ uptake in yeast with the triple *mep1/2/3* mutant [16]. Subsequently, a number of *AMTs* was identified in *A. thaliana*, rice (*Oryza sativa*), tomato (*Solanum lycopersicum*), and many other species of angiosperms [5,17,18]. The constructed *AMT* phylogenetic tree showed that the family members can be divided into clades (*AMT1s* and *AMT2s*) [7,19], with fewer members in *AMT2*, which are more homologous with the *AmtB* of *Escherichia coli* and *Meps* of yeast (*Saccharomyces cerevisiae*) [20,21]. Previous studies showed that plant *AMTs* are mainly located in the cell membrane with 7–12 transmembrane domains (TMDs) [22]. In *A. thaliana*, six *AtAMT*s were identified, of which five belong to the *AMT1* subfamily (i.e., *AtAMT1.1~AMT1.5*) and only one (*AtAMT2*) belongs to that of *AMT2*. More specifically, *AtAMT1.2*, *AtAMT1.3*, *AtAMT1.5*, and *AtAMT2* were found to be mainly expressed in roots. *AtAMT1.4* showed pollen-specific expression, and *AtAMT1.1* was found to be highly expressed in roots, stems, and leaves [5]. Characterization of the Arabidopsis mutants demonstrated that three members of the *AtAMT1* subfamily (*AtAMT1.1*~*AtAMT1.3*) belonging to the HATs contributed about 90% of the NH_4_^+^ uptake [10,23]. In rice, NH_4_^+^ transporters are more abundant and have diverse expression patterns compared to those in Arabidopsis due to waterlogging tolerance [15,24,25]. Previous reports showed that the expression of *AMT*s was regulated by light, phytohormones, and environmental stresses [18,26,27] in the model plant as well as in major crops.

Quinoa is an Andean allotetraploid crop with excellent nutritional quality due to its high protein content, balanced amino acid composition, good fatty acid profile, vitamins, mineral elements, dietary fiber, and gluten-free seeds [28]. The protein content of quinoa grain ranges from 12% to 23% [29], with higher concentrations of nine essential amino acids [30]. The quality and yield of quinoa, especially the protein and amino acid contents in the seeds, are mainly dependent on the macromineral nutrients in which nitrogen is an important player.

Quinoa grows in nitrate-dominated drylands. The whole growth and development of quinoa with the largest nutrition demand occurs with the panicle appearance at the top (with 11–16 whorled leaves) [31]. Application of nitrogen fertilizer will lead to maximized agricultural yields, and NH_4_^+^-dominated fertilizer application in soils as well as the top parts of the quinoa in the largest nutrition demand stage will be better agronomical practice in terms of quinoa cultivation to meet the nitrogen demand, in which quinoa has an efficient uptake ability from soils or leaf surfaces following the assimilation to accumulate organic nitrogen metabolites (e.g., nucleotides and amino acids) for better panicle growth and development. Previous studies showed that when supplied with equivalent concentrations of nitrogen, NH_4_^+^ can support similar or greater quinoa growth and yields compared with nitrate [32,33], indicating that quinoa has a strong NH_4_^+^ absorption and assimilation system. The ammonium uptake in plants is controlled by *AMTs* [34]. Although *AMTs* and their expressions have been reported in many plants, the *AMT* gene family, functions, and regulations remain unknown in quinoa despite the current availability of large omics data concerning *Chenopodium quinoa* [35]. In this study, the quinoa *CqAMT* family, the function of gene members, and their regulatory networks were investigated based on the multi-omic strategy and molecular methods for better understanding the NH_4_^+^ uptake, assimilation, and redistribution in quinoa for further improvement in NUE as well as optimizing cultivation for both yield and quality of quinoa production.

## 2. Results

### 2.1. Identification of AMT Genes from Chenopodium quinoa and Chromosome Localization

After removing redundant entries, 12 *AMT* genes were identified from the quinoa genome database using both BLAST and hmmer searching. These genes were located on different chromosomes (Chrs) of *Chenopodium quinoa* and were named according to their homologous gene class (Table 1, Figure 1 and Appendix A). Chrs 5, 7, and 12 had three *CqAMT*s, while Chrs 11, 17, and 18 had only one copy of *CqAMT* (Appendix A); no *CqAMTs* were found on Chrs 1, 2, 3, 4, 6, 8, 9, 10, 13, 14, 15, and 16. Some *CqATMs* were located at the ends of some Chrs; for example, *CqAMT2.2b*, *CqAMT1.4a*, and *CqAMT1.3b* were located on the end parts of Chrs 11, 17, and 18, respectively. The ORF lengths of the *CqAMTs* ranged from 933 to 1518 bp. The protein sequences of *CqAMTs* comprised 310~505 amino acids (Aas) with predicted molecular weights from 33.79 (*CqAMT3.1a*) to 54.12 kDa (*CqAMT2.2a*). The range of the predicted isoelectric points (pIs) of the *CqAMTs* was 5.47~9.52, with 7~11 conserved TMDs; all 12 *CqAMTs* were predicted to be located on the plasma membrane (Table 1 and Appendix A).

### 2.2. Phylogenetic and Evolution Analysis of CqAMTs

In plants, *AMTs* are divided into two gene subfamilies, *AMT1* and *AMT2* [7]. To reveal the evolutionary relationships in the *CqAMT* family, a phylogenetic tree was constructed with 51 *AMTs* from 6 different plant species, namely quinoa (*Chenopodium quinoa*), Arabidopsis (*A. thaliana*), rice (*O. sativa*), tomato (*Solanum lycopersicum*), tea (*C. sinensis*), and poplar (*P. trichocarpa*), as listed in Appendix A. The results of a non-rooted cluster tree showed that the AMT proteins could be divided into three groups: Group I, Group II, and Group III; the AMT1 subfamily was divided into Group I and Group III (Figure 1). Group I comprised 5 members (i.e., *CqAMT1.3a*, *CqAMT1.3b*, *SlAMT1.3*, *CsAMT1.3a*, and *CsAMT1.3b*), with 2 *CqAMTs* from quinoa, whereas Group III contained 26 members, including 8 *CqAMTs* (i.e., *CqAMT1.2a*, *CqAMT1.2b*, *CqAMT1.2c*, *CqAMT1.2d*, *CqAMT1.4a*, *CqAMT1.4b*, *CqAMT2.2a*, and *CqAMT2.2b*). *CqAMT2.2a/b* was grouped into a small clade with *PtrAMT2.2* sistered with another one, composited with *SlAMT1.1* and *CsAMT1.1a/b/c*. *CqAMT1.4a*/*b* was found to have a highly shared homology with *CsAMT1.4* and *AtAMT1.4*, clustered into a small clade located in the middle part of Group III. *CqAMT1.2a*/*b* and *CqAMT1.2c/d* were clustered together as another independent small clade located in the base part of Group III. Group II comprised members of the *AMT2* subfamily and had two *CqAMTs* (*CqAMT3.1a* and *CqAMT3.1b*) and many other *AMT*2 subfamily members from selected species in this study (Appendix A); here, *CqAMT3.1a* and *CqAMT3.1b* were found to have a highly shared homology with *CsAMT3.1* and *OsAMT3.1* (Figure 1A). Furthermore, there were six gene pairs of *CqAMTs* (i.e., *CqAMT2.2a*/*b*, *CqAMT1.4a*/*b*, *CqAMT1.2a*/*b*, *CqAMT1.2c*/*d*, *CqAMT1.3a*/*b*, and *CqAMT3.1a*/*b*) that were predicted to be in quinoa; their presence suggested that *CqAMT* pairs of tetraploid quinoa may be derived from the progenies of the A genome and the B genome of the diploid species, respectively [36]. Thus, based on the identified 12 *CqAMT*s in quinoa, we identified 5 *AMT* homologous genes in both the genome of *Chenopodium watsonii* (*Cw*), endemic in North America, and the genome of *Chenopodium suecicum* (*Csu*), endemic in Eurasia. This was achieved through homolog searching (Appendix A) for *Cw* (A genome) and *Csu* (B genome) diploid species; these had higher homology with the A sub-genome and the B sub-genome of quinoa, respectively. The collinear analyses conducted using MCScanX showed that homologous *CqAMT* gene pairs had one vs. one or one vs. more homologous relationships between *Cw* and quinoa as well as *Csu* and quinoa, respectively. Interestingly, *CqAMT3.1* was only located on the colinear region between quinoa and *Cw* with homologous genes of *Cw009490*; meanwhile, *CqAMT1.2a*, which has a 100% protein similarity with Csu016931 (*CsuAMT1.2a*) and 82.2% similarity with Cw010202 of *Cw*, was not located in the collinear regions between quinoa and the diploid species of both the A and B genomes (Figure 1B; Appendix A). This indicates that the presence of *CqAMT1.2a* in the tetraploid species quinoa may be due to gene duplication or chromosome arrangement, which occurs after tetraploidization during domestication. It can also be attributed to the adaptive evolution of quinoa for survival in specific environmental niches.

### 2.3. The Structural Features of CqAMTs

Ten conserved motifs were identified in *CqAMTs* using MEME (Appendix A and Figure 2B); the conserved amino acid sequences of the motifs and their functions were further predicted using the Pfam database. The results showed that the length of the conserved motifs’ amino acid residues ranged from 21 to 50. Seven motifs (motifs 1~7) were annotated to have an *AMT* function, while three motifs (motifs 8~10) were predicted to have an unknown function (Appendix A, Figure 2). Interestingly, motifs 7 and 9 were found to be highly conserved in all *CqAMTs*, whereas the other eight motifs (i.e., motifs 1, 2, 3, 4, 5, 6, 8, and 10) were present only in the *AMT1* subfamily (Figure 2A). Members of the *AMT2* subfamily had only the conserved motifs 7 and 9 (Figure 2A). We found that the *AMT1* subfamily contained 10 motifs; these are with the exception of *CqAMT1.2c*, which did not have motifs 4 and 9, and *CqAMT1.3a*, which did not have motif 2. Motif 4 was located at the N terminus of the proteins of the *AMT1* subfamily, with the exception of *CqAMT1.2c*. Meanwhile, motif 3 was located at the end of all the proteins of the *AMT1* subfamilies (Figure 2B).

Different compositions of introns and exons have been reported to lead to changes in gene function [37]. To reveal the gene structure of *CqAMTs*, the distribution of exon–intron combinations was analyzed. As shown in Figure 2C, four genes (*CqAMT1.2a*, *CqAMT1.2b CqAMT1.2d*, and *CqAMT1.4b*) had no introns; seven *CqAMT* genes (*CqAMT 3.1a*, *CqAMT1.2c*, *CqAMT2.2a*, *CqAMT2.2b*, *CqAMT1.3a*, *CqAMT1.3b*, and *CqAMT1.4a*) contained two exons separated by one intron. Only *CqAMT3.1b* had two introns and three exons (Figure 2C). Based on our RNAseq data, all 12 *CqAMTs* were detected to have full CDS; they were also found to have the flanking 5′ and 3′ un-translational region (UTR) in the first and last exons, respectively (Figure 2C).

### 2.4. Prediction of Cis-Elements in the Promoters of CqAMTs

The characterization of the promoter regions is essential in understanding the potential transcriptional regulation of these *CqAMTs*. The 2000 bp upstream sequences from the translation start site (*ATG*) of the *CqAMTs* were extracted for the cis-element analyses performed through Plant CARE. The results showed that 18 types of cis-elements were predicted; these include AT-rich DNA binding, MYB binding, zein binding, defense- and stress-responsive elements, phytohormone-responsive elements (e.g., abscisic acid (ABA), auxin (AUX), salicylic acid (SA), and gibberellin (GA)), environment-responsive elements (e.g., anaerobic, light, low-temperature, anoxic specific, and elicitor-mediated activation), and circadian-mediating elements. These cis-elements were widely and randomly distributed in the promoter regions; light responsiveness-related cis-elements (LREs) appeared most frequently. Interestingly, all *CqAMTs* had LREs in which *CqAMT1.2a* and *CqAMT1.4b* had six LREs in their promoter, indicating that light is a major environmental factor that regulates NH_4_^+^ transportation and assimilation. Many stress-related elements and phytohormone-responsive factors’ binding sites were found on the promoter of *CqAMTs*; this indicated that *CqAMTs* may play vital roles in the response to abiotic stress in quinoa as well as endogenous cues related to plant growth and development (Figure 3). Here, various TFs, such as MYB, bHLH, zinc fingers (ZFs), WRKY, B3 type, NLPs, G2-like, and NF-Y, may be involved in the regulation of *CqAMTs*.

### 2.5. Expression Analysis of CqAMTs Under Different NH_4_^+^–Nitrogen Concentrations in Hydroponic Cultivated W32 and Different Tissues of Field-Planted W19 and W25

The RNAseq data of the leaves and roots of W32 quinoa planted under different NH_4_^+^–nitrogen concentrations (0, 8, and 21 mM NH_4_^+^ only) in modified Hoagland solution and various tissues of W19 and W25 cultivated in field conditions, which are highland ecotype germplasms, were analyzed, focusing on the expression of *CqAMTs* in the following organs: roots (Rs); leaves (Ls) for seedlings (representative vegetative phase) of W32; leaves (Ls); and panicles (Ps) at both flowering (F) and seed-filling (S) stages (representative reproductive phase) of W19 and W25 planted in the autumn–winter season. The statistical results of the Illumina RNAseq data showed that short reads (fastq) had more than 91.86% Q30 and 24 M reads/per sample (Appendix A). The expression levels of the genes were calculated as TPM. *CqAMT3.1a* and *CqAMT3.1b* (belonging to *AMT2* subfamily) and *CqAMT2.2a*, *CqAMT2.2b*, *CqAMT1.3a*, and *CqAMT1.3b* (belonging to *AMT1* subfamily) had higher expressions in the Ls than in the Rs of W32, indicating their main function on the leaf or aerial part of quinoa NH_4_^+^ transportation. Meanwhile, *CqAMT1.2c* expression showed the opposite trend; it had lower expressions in the Ls but higher expressions in the Rs. This indicates that the main function in the roots for NH_4_^+^ transportation. *CqAMT1.2a*, *CqAMT1.2b*, and *CqAMT1.2d* have the same expression patterns, with high expression in both the Rs and the Ls; this is especially the case in nitrogen-deficient roots and leaves in the vegetative phase of quinoa. Interestingly, four *CqAMTs* (i.e., *CqAMT2.2a*, *CqAMT2.2b*, *CqAMT1.4b*, and *CqAMT1.4a*) belonging to the *AMT1* subfamily had an overall lower expression in all the detected organs compared with other *CqAMTs* (Figure 4); here, the expression of *CqAMT1.4a* was very low in both the leaves and roots of W32 at any NH_4_^+^–nitrogen concentration. Meanwhile, *CqAMT1.4b* had relatively increased expression in roots in comparison with that of *CqAMT1.4a* in the vegetative phase in hydroponic-cultivated W32. *CqAMT2.2a* and *CqAMT2.2b* had similar expression patterns, with increased expression in the Ls in comparison with the Rs. *CqAMT2.2a* had an overall high expression in comparison with *CqAMT2.2b* in the reproductive phase of the field-cultivated condition. In both the leaves and panicles of W19 and W25, *CqAMT2.2b*, *CqAMT1.4a*, and *CqAMT1.4b* were detected with lower expression, while other *CqAMTs* had relatively high expression. The results also showed that *CqAMT1.2b*, *CqAMT1.3a*, *CqAMT1.2a*, *CqAMT1.3b*, *CqAMT1.2d*, *CqAMT3.1a*, and *CqAMT3.1b* had higher expression in Ps at the flowering stage of both W19 and W25; only *CqAMT1.2d* and *CqAMT1.2b* had a high expression in the Ps at the seed-filling stage of both W19 and W25; by contrast, *CqAMT1.3a*, *CqAMT1.2a*, *CqAMT1.3b*, and *CqAMT3.1b* had higher expressions in the Ps at the seed-filling stage of W19. *CqAMT1.4a* and *CqAMT1.4b* were also detected, with relatively high expressions at the flowering and seed-filling stages in the Ps of W25. These results indicate their role of NH_4_^+^ transportaion in the flowering and seed development of quinoa (Figure 4B). To further validate the gene expression profiles obtained from RNASeq data, the expressions of *CqAMT1.2a*, *CqAMT1.2b*, *CqAMT1.2c*, *CqAMT1.2d*, *CqAMT1.3a*, *CqAMT1.3b*, *CqAMT1.4b*, *CqAMT2.2a*, *CqAMT3.1a*, and *CqAMT3.1b* in the Rs and Ls of W32 cultivated under different NH_4_^+^ concentrations by qRT-PCR were analyzed. *CqAMTs* expression detected by qRT-PCR was found to be relatively consistent with their expressions detected by transcriptome data (Figure 5). The profiles of *CqAMT* expression revealed the diverse functions of the *CqAMT* family in the growth and development of quinoa by NH_4_^+^ absorption, transfer, and redistribution.

### 2.6. Confirmation of CqAMTs NH_4_^+^ Uptake Function and Subcellular Localization of CqAMT1.2a

To characterize the function of *CqAMTs* in NH_4_^+^ transport, the CDSs of 11 *CqAMTs* (except *CqAMT2.2b*, the full coding sequence (CDS) of which was not successfully amplified even when it was repeated several times) were cloned into pYES2; then they were introduced into the yeast strain 31019b (mep1Δ, mep2Δ: LEU2, and mep3Δ: KanMX2 and Δura3). Triple *mep1/2/3*-mutated yeast strain 31019b cannot grow at a NH_4_^+^ concentration below 5 mM [38]. The results showed that the control transformants carrying the empty vector pYES2 could not grow on a medium with low NH_4_^+^ concentration (less than 3 mM); however, the yeast transformants harboring any of the 11 *CqAMTs* restored the growth of yeast on 3 mM NH_4_^+^ and 0.3 mM NH_4_^+^ (Figure 6A). These results indicated that all the cloned full CDSs of 11 *CqAMTs* mediated NH_4_^+^ permeation across the plasma membranes of the yeast cells. Most of the members of the *AMT1* subfamily had 7~11 transmembrane domains (TMDs), with 11 TDMs found to be dominant in quinoa (Table 1). *CqAMT1.2a,* which had 11 TDMs, with higher expressions in all organs of W19, W25, and W32, was chosen for its representativeness in subcellular location detection. The results showed that the *CqAMT1.2a* that had fused with GFP were transiently expressed in tobacco leaves; the distinct fluorescence of CqAMT1.2a:GFP was observed in membrane systems, especially in the cell membrane. This indicated that *CqAMT1.2a* is mainly localized at the cell membrane (Figure 6B). The other 11 *CqAMT* genes were also predicted to be located in the plasma membrane (Table 1), suggesting that all the detected *CqAMT* members were located on membranes, especially plasma membranes.

### 2.7. Gene Regulatory Networks Are Related to Nitrogen Absorption and Assimilation

Nitrogen is a necessary macronutrient for plant primary and secondary metabolism; it is also a signaling molecule for the regulation of plant growth and development. To analyze the gene regulatory networks, especially the major regulators for NH_4_^+^ transportation and assimilation, WGCNA was used to construct a gene co-expression module using all the RNAseq-based gene expression matrices of TPM (TPM matrix). Notably, 95 gene modules (GMs), labeled with different colors, were identified (Appendix A–C). Based on the GMs, 8 out of 12 *CqAMTs* were grouped into 4 GMs, in which *CqAMT2.2a*, *CqAMT3.1b*, *CqAMT1.2c*, and *CqAMT1.2d* were clustered into the turquoise GM (TGM); *CqAMT1.2a* and *CqAMT1.2b* were grouped into blue GM (BGM); *CqAMT3.1a* was grouped into the light-green GM (LGM); *CqAMT1.3b* was grouped into the pale-turquoise GM (PGM) (Appendix A). A total of 247 genes existed in the PGM gene set (Appendix A), among which 8 TF-encoding genes were detected. However, except *CqAMT1.3b*, no other genes associated with nitrogen transportation and assimilation were found in PGM; none of the detected TFs in PGM have binding sites on the promoters of *CqAMT1.3b (AUR62020119)*. Similarly, a total of 432 genes existed in the LGM gene set, among which 15 TF-encoding genes were detected (Appendix A). Further analyses of nitrogen transportation and assimilation genes in LGM found that only *CqAMT3.1a* (*AUR62039048*) and a glutamine synthetase (GS) gene (*AUR62026959*) were detected. However, further analysis showed that the promoters of both *CqAMT3.1a* and *GS (AUR62026959)* also had no binding sites for the TFs detected in the LGM. Therefore, the analyses of nitrogen regulatory networks were focused on the BGM, with 3299 genes (Appendix A). Among these, two *CqAMT*s and five GS-coding genes were identified (Appendix A). We also investigated the TGM, with 4332 genes (Appendix A), in which 4 *CqAMTs*, 1 *GS,* and 9 nitrate transporters (*NRTs*) were identified (Appendix A). Correlated expressions were observed between *CqAMT*s and other nitrogen assimilation genes and TFs in the same module (absolute R > 0.6 and *p*-value < 0.05). Furthermore, the TF binding site(s) existed in the promoters of those target genes were detected (Appendix A). Notably, 35 and 25 putative TF genes associated with the regulation of nitrogen transportation and metabolism were initially identified in the BGM (Appendix A) and the TGM (Appendix A), respectively, with their expression patterns being demonstrated by heatmaps, along with nitrogen transportation and assimilation genes in both modules (Appendix A). To further narrow down the TFs scope, especially in the regulation of the root system of hydroponically cultivated W32, measurements were taken to analyze the changes in phenotypes and enzyme activities involved in nitrogen assimilation. Significant changes were found in the phenotypes and physiological traits of quinoa roots with treatments using different NH_4_^+^ concentrations (Appendix A). Root length and the root surface area were significantly increased under nitrogen-deficient conditions, whereas the number of root tips reached their maximum in the eight mM NH_4_^+^ concentration, and the average diameter of the roots showed a similar trend (Appendix A–D). The enzymatic activities associated with nitrogen assimilation, including the NR, GS, and NiR in roots, declined under low-N stress (Appendix A–H). The dry weight of the quinoa root reached the maximum value when nitrogen was deficient and gradually decreased with the increase in nitrogen concentration. By contrast, the fresh weight of quinoa root reached a maximum value when the nitrogen concentration was 8 mM, but there was no significant difference compared with 0 mM and 21 mM NH_4_^+^ (Appendix A). Collectively, these results implied that low-N stress induces key gene expression changes, thus resulting in changes in the physiological and phenotypical traits of quinoa roots. The correlations between TF expressions and root phenotypes (i.e., fresh weight, dry weight, specific length, surface area, average diameter, and number of root tips) and enzyme activities associated with nitrogen assimilation in roots, namely GOGAT, GS, NiR, and NR (Appendix A), were analyzed. Filtering the results considering the standard of absolute R-value > 0.6 and *p*-value < 0.05, the negative or positive correlations between at least three types of the above root phenotypes and the activities of three kinds of enzymes in roots of quinoa were investigated; thus, the core regulatory TFs were finally selected: 15 TFs in the BGM and 21 TFs in the TGM (Appendix A highlighted in red). The final 15 TFs in the BGM were grouped into 9 TF families (i.e., bHLH, bZIP, C2H2, G2-like, GRAS, NAC, NF-Y, Trihelix, and WRKY), while 21 TFs in the TGM were grouped into 10 TF families (i.e., ARR-B, B3, bZIP, C3H, GRAS, HSF, NAC, NF-Y, Nin-like (NLP), and WRKY) (Figure 7A–D). Additionally, in the BGM, *AUR62032797(G2-like)* and *AUR62007169(Trihelix)* were found to be negatively regulated TFs; this was indicated by the negative correlation observed between their expression and the expressions of the *CqAMT1.2a* and *GS* genes (i.e., *AUR62041056* and *AUR62037667*). Specifically, *AUR62032797(G2-like)* expression was negatively correlated with *CqAMT1.2b* and *GS* (*AUR62041382*) expression, while that of *AUR62007169(Trihelix)* was negatively correlated with *AUR62019375(GS)* expression. The positive regulatory TFs in the BGM, indicated by TF expressions (e.g., *G2-likes* (*AUR62024357* and *AUR62025961*), *AUR62005168(NF-YB)*, *AUR62038058(NF-YC)*, *AUR62011806 (bZIP)*, *AUR62031178 (bHLH)*, *AUR62038383 (C2H2),* and *AUR62020401 (WRKY)*) were positively correlated with the expression of *CqAMT1.2a/b* and *GSs* as well as the activities of GOGAT, GS, NiR, and NR in roots. In the TGM, the expression of the negatively regulated TFs of *AUR62000210(HSF)* and *AUR62011826(B3)* had negative correlations with the expression of most *NRTs* and *GS (AUR62017693)*. *CqAMT2.2a* was negatively regulated by *AUR62011973(HSF)*, *AUR62011826(B3)*, *AUR62017780(C3H)*, *AUR62033188(NF-YB)*, and *AUR62023070(NF-YA)* but positively regulated by *AUR62035070(ARR-B)* and *AUR62000210(HSF)*. The expressions of *NLPs* (*AUR62012917* and *AUR62040813*) and their possible regulated downstream TFs of *AUR62023070(NF-YA)* and *AUR62033188 (NF-YB)* (the expressions of which were positively correlated with those of *CqNRTs* (e.g., *AUR62001496*, *AUR62009875*, *AUR62001495*, and *AUR62009874*)) were negatively correlated with *CqAMT2.2a* expression, as well as the activities of GOGAT, GS, NiR, and NR in the roots (Figure 7E). The results suggested that the abovementioned TFs may be mainly involved in the regulation of nitrogen transportation and the assimilation of quinoa, especially in the roots. For the confirmation of TF expression detected by RNAseq, *AUR62000210(HSF)*, *AUR62005714(ERF)*, *AUR62009548(GRAS)*, *AUR62011-826(B3)*, *AUR62012917(Nin−like)*, *AUR62018123(WRKY)*, *AUR62036595 (bHLH)*, *AUR6203-6626(NAC),* and *AUR62040941 (C2H2)* were chosen for qRT-PCR detection with the primers of which are listed in Appendix A. Gene expression patterns exhibited nearly consistent trends in both methods (Appendix A). Further prediction of TF binding sites (TFBSs) was conducted for the promoters of the genes associated with nitrogen transportation and assimilation; putative TFs were detected in both BGM and TGM (Appendix A). Notably, ~24 kinds of cis-elements were found, including 139 ABREs (involved in the abscisic acid responsiveness), 289 MYCs (motifs responding to chilling and phytochromes), 163 STREs (stress response elements), 90 CGTCA motifs (involved in the MeJA responsiveness), 90 as-1 (salicylic acid- and auxin-responsive element), 90 TGACG motifs (involved in the MeJA responsiveness), and 145 AREs (antioxidant response element related to the anaerobic environment). In addition to the abovementioned cis-elements, the ERE, WUN-motif, and GARE-motif binding sites were also predicted to be present on the promoters of those genes (Appendix A and Figure 7). Based on these results, the gene regulatory networks in the BGM and TGM, including TFs, are involved in targeting the genes of nitrogen transportation, and assimilation was predicted(Figure 7E,F); additionally, TFs, which are the possible target of TF genes (Figure 7G,H), were also constructed with a CytoHubba plug-in using the MCC algorithm in Cytoscape. The outputs are visualized in Figure 7. Furthermore, the putative TFs were further referred through phylogenetics, with reference to the homologous TFs, whose functions were confirmed in *A. thaliana* in addition to some other species (Appendix A). Based on the above analyses, a comprehensive theory regarding nitrogen transportation, assimilation, and their regulation for growth and development in quinoa roots was hypothesized, the summary of which is illustrated in Figure 8.

## 3. Discussion

Quinoa, with gluten-free seeds, has a high content of proteins and essential amino acids [42]; additionally, it has a complete nutritional profile, meeting the nutritional requirements for humans [43,44,45]. The yields and quality of quinoa are strongly dependent on the supply of nitrogen; NH_4_^+^ is one of the main forms absorbed by plants [46]. Therefore, a series of NH_4_^+^ transportation, assimilation, and regulatory genes exist in plants, which play an important role in maintaining the dynamic balance of nitrogen and ensuring the proper utilization of nitrogen for plant growth and development. *AMTs* for NH_4_^+^ transport have been identified in many plants (e.g., Arabidopsis and rice), and some studies have investigated their assimilation process using GS and GOGAT [47,48]. The members of the *AMT* family have been reported to vary significantly among plant species. For example, 6, 10, 3, 5, 7, and 16 *AMTs* were identified in *A. thaliana*, *O. sativa*, *S. lycopersicum*, *N. tabacum*, *P. trichocarpa,* and *Camellia sinensis*, respectively. In this study, 12 putative *CqAMTs* with 6 gene pairs (e.g., *CqAMT1.2a/b, CqAMT1.2c/d, CqAMT1.3a/b*, *CqAMT1.4a/b*, *CqAMT1.4a/b*, *CqAMT2.2a/b*, and *CqAMT3.1a/b*) were identified, each having different expressions in the different organs and developmental stages of quinoa (Figure 1, Figure 2, Figure 4 and Figure 8). All CqAMT proteins were predicted to harbor the NH_4_^+^ transport domain (PF00909) with the conserved motifs of NH_4_^+^. TMDs were varied in the different members of *CqAMTs* (Figure 2, Table 1) [49]. Interestingly, *CqAMT3.1a*/b only contained two conserved motifs (i.e., motifs 7 and 9), with 7 and 11 TMDs, respectively (Table 1, Figure 2B). *CqAMT3.1a/b,* grouped into a clade with *OsAMT3.1* and *CsAMT3.1* in Group II (Figure 1), belongs to the LAT of the *AMT* 2 subfamily [50]. *CqAMT3.1a/b* had similar expression patterns, with the previously reported *CsAMT3.1* of *Camellia sinensis* L and *OsAMT3.1* of rice [51] showing low expression in nitrogen (N)-supplied roots but strong expression in leaves [52,53] (Figure 4A and Figure 8). These findings suggest that *CqAMT3.1a/b* may work on the aerial part of quinoa, mainly in the leaves and panicles, especially around the plant’s reproductive development. The phylogenetic relationships of the remaining five gene pairs of *CqAMTs*, with more conserved motifs in the protein sequence, were clustered into Groups I and III. These belonged to the HAT of the *AMT1* subfamily (Figure 1A) [27,53]. *CqAMT1.3a/b,* belonging to Group I, had higher expression in the leaves but low expression in the roots, with a similar expression pattern as *CqAMT3.1a/b*; this suggests that they are redundant members of the *AMT1* subfamily, with the same functions as those previously described for the *CqAMT3.1a/b* of the *AMT2* subfamilies in quinoa. In Arabidopsis, *AtAMT1.4* was found to be highly expressed in pollen, contributing to pollen development [8]. *CqAMT1.4a/b* was grouped into a clade with *AtAMT1.4* in Group III (Figure 1A); this had a relatively lower expression; however, in the panicles during the reproductive stage, it had a higher expression than that of leaves and roots (Figure 4B). This suggested that *CqAMT1.4a/b* may be involved in NH_4_^+^ transportation in the flowers and pollen development of quinoa, with a similar function of *AtAMT1.4* in Arabidopsis. *CqAMT1.2a/b* and *CqAMT1.2c/d* were clustered into an independent clade in the base part of Group III; here, *CqAMT1.2a/b/d* had no intron, with a higher expression in the Ls and the Rs but a low expression in the Ps: Ps < Ls < Rs. Meanwhile, *CqAMT1.2c* with a intron had a higher expression only in the Rs: Rs > Ps > Ls. This suggested that the intron of *CqAMT1.2c* may play a regulatory role for *CqAMT1.2c* expression and RNA stability. Similarly, different introns resulted in different expressions, which was also observed in the gene pairs of *CqAMT1.4a/b* and *CaAMT3.1a/b* (Figure 2 and Figure 4). These are consistent with previous reports that introns are usually involved in the regulation of gene expression or RNA stability [54,55]. *CqAMT2.2a/b* had a lower expression in the Rs but a higher expression in the Ls and the Ps, especially in the reproductive stage. This suggested that they may also play important roles in fruit and seed development and may finally contribute to the yields and overall protein content in quinoa seeds [56,57]. Of the 12 *CqAMT* genes, 11 (with the only exception of *CqAMT2.2b*) were confirmed to be involved in NH_4_^+^ uptake during the triple mutation of *mep1/2/3* in yeast (Figure 6). Although plants and yeast have different ammonium transportation and assimilation systems, previous studies have rseported that the GS/GOGAT pathway plays a role in glutamate synthesis and nitrogen assimilation. This is especially the case for the key players: glutamate dehydrogenase (GDH) enzyme-coding genes have a mutated strain (*gdh1Δ2Δ3Δ*) in *Saccharomyces cerevisiae* [58]. The redundant ammonium assimilation pathway (i.e., GDH, GS/GOGAT, and proline utilization (PUT)) [59] ensures quick ammonium–glutamate conversion in yeast. This partially explains the way in which the 11 *CqAMTs* resulted in mutant yeasts without significant differences in growth status, i.e., through either HAT- or LAT-type transformation (Figure 6). The confirmation of the complementary role of ammonium uptake among the 11 tested *CqAMTs* in the *mep1/2/3*-mutated strains provides evidence for the ammonium transportation function of *CqAMTs*. Their specific role(s) in the different organs and tissues of quinoa for ammonium absorption and/or translocation need further investigation. Compared with *AtAMTs* in Arabidopsis, quinoa was found to have more *AMT* members (nearly doubled). This suggests that they may be derived from the hybridization of ancestral diploids. The genome sequences of the two most possible candidates for their extant, ancestral-related diploids, *Cw* and *Csu*, contain higher homologous DNA sequences in the quinoa genome [35], with five *AMT* genes in each species (Figure 1); this suggests that quinoa might have undergone genome-wide replication through polyploidy events [35]. This theory is consistent with the evidence of intron changes in the gene pairs of *CqAMT1.4a/b*, *CqAMT1.2c/d*, and *CqAMT3.1a/b* (Figure 2).

In general, *AMTs* are widely distributed in plants and expressed in multiple tissues. There is a clear scope for the specific physiological roles of the distinct homologs, located in different cell types or tissues [49]. To investigate the functions and regulation of *CqAMTs* in the growth and development of quinoa, the cis-element distribution in *CqAMT* promoters was characterized, as well as their expression patterns in the different tissues obtained via RNAseq analysis (Figure 4). This was followed by qRT-PCR quantification (Figure 5). *CqAMT3.1b*, *CqAMT1.2c*, *CqAMT1.2b*, *CqAMT2.2a*, *CqAMT 1.3a*, *CqAMT1.3b*, *CqAMT1.4a*, and *CqAMT1.4b* in quinoa were identified to have anaerobiosis-inducible promoter regions (Figure 3). This suggests that the root absorption and assimilation of NH_4_^+^ in the soil through these *CqAMTs* particularly occur in low-oxygen conditions [60,61]. The higher expression of many *CqAMTs* (Figure 5) in leaves suggests an association with NH_4_^+^ transport and assimilation in the aerial parts of quinoa; this is consistent with the gene promoters having many light-responsive elements. The results suggest that these *CqAMTs* might participate in the regulation of photorespiratory NH_4_^+^ metabolism as well as carbon and N assimilation [62]. However, the details of this mechanism need further investigation. Remarkably, the predicted promoters of *CqAMT2.2a* and *CqAMT1.2b* contained cis-elements associated with defense and stress response [63,64]. This suggests that the function and regulation of these genes may also be related to stress induction for specific ecological adaptation. *CqAMT1.2d* was found to be highly expressed in the Ps; their promoter regions were found to have cis-elements that were responsive to phytohormones (e.g., ABA and GA) associated with flowering and seed development (Figure 3 and Figure 4B). This suggests that *CqAMT1.2d* may contribute to the N supply in the reproductive phase, especially in the mature development of quinoa seeds. Plants mainly absorb water and nutrients from the soil through the roots; the specific root length and surface area are commonly used indexes for evaluating nutrient absorption efficiency. In this study, we analyzed the physiological and phenotypical traits of quinoa roots under different ammonium N concentrations. The results showed that the specific root length and surface area significantly increased under low N (Appendix A). Root nutrient absorption depends on a limited number of binding sites on the root tips and soil particles, thereby increasing nutrient availability and the proportion of total nutrient uptake from the soil by the roots, resulting in a larger root length and root surface area under low-N or N-deficient states [65,66]. These results are consistent with our current root phenotypic observations of hydroponically cultivated W32; this approach allowed us to easily detect phenotypic changes in the root system. Our findings showed that the lengths, area, and dry weight of the roots were negatively correlated with NH_4_^+^ concentration (Appendix A).

The biochemically usable state of N is NH_4_^+^. Aside from the direct absorption of NH_4_^+^, another way to ensure the presence of NH_4_^+^ in a plant is through NO_3_^−^ uptake and its reduction to NH_4_^+^. The transportation and assimilation process of NO_3_^−^ in plants is initiated by the transportation of nitrate by cell membrane transporters (NRTs). When nitrate enters into the cell, the reduction of nitrate into nitrite with either NADH or NADPH in the cytosol by a two-electron donor reaction through nitrate reductases (NRs) is the first step of the process [67]. Nitrite is a highly toxic ion; therefore, plant cells immediately transport the nitrite into chloroplasts in the aerial parts of a plant or through the plastids in the roots, decreasing the harmful effects of toxic ions. After entering these organelles, the enzyme nitrite reductase (NiR) catalyzes a reduction reaction of nitrite into NH_4_^+^ through a six-electron reduction process.

There are two assimilation paths considering organic N: (i) NH_4_^+^ and glutamate are transformed into glutamine, catalyzed by GS. Then, the resulting glutamine and alpha-ketoglutaric acid (2OG) form two glutamates, catalyzed by GOGAT. Here, part of the glutamate is involved in the synthesis of the other amino acids and nucleotides. The remaining glutamate re-enters the GS/GOGAT cycle. (ii) NH_4_^+^ is directly catalyzed by glutamate dehydrogenase and produces a condensation reaction with 2OG to synthesize glutamate [68]. In these metabolic pathways, ammonium is either reduced in the conversion from NO_3_^−^ or directly provided by *AMTs*. In the current study, the four ammonium transporter (*CqAMT1.2a/b/c/d)* genes were highly expressed in the roots; the transcription factors (TFs) *AUR62011806 (bZIP)* and *AUR62007613 (bZIP)* may have positively controlled their expressions. This is evidenced by their positive expression correlation with the abovementioned *CqAMT1.2a/b/c/d*. In contrast, *AUR62005168 (NF-YB)*, *AUR62020401 (WRKY)*, *AUR62024357 (G2-like)*, and *AUR62025961 (G2-like)* have negative regulatory function. This is evidenced by the negative correlation observed between their expression and the ammonium transporter genes. However, these predicted regulatory associations need to be further verified in future studies (Figure 7 and Figure 8). The genes identified in this study were highly expressed in the roots and panicles; in particular, either encoding *CqAMTs* or in their regulatory TFs laid a foundation for breeding quinoa varieties with improvements in NUE for increasing crop yield with reduced fertilizer supply [69,70,71,72]. In major crops such as wheat, researchers have identified GS and GOGATs through comparative genomic approaches; these have been found to be important candidates for improving N metabolism and N deficiency resistance [39]. A large number of TFs also have been identified to regulate the expression of N-related genes in rice [73]. Recent study results have revealed that the overexpression of *OsMYB305* in rice enhances N uptake under low-N conditions [74]; meanwhile, *GLK5* and *bZIP108* genes have been found to be involved in regulating genes in response to N deficiency [75]. Similarly, ERF and WRKY TFs have been reported to preferentially regulate the expression of N-related genes [76]; *OsNLP4* plays a pivotal role in NUE [77]. A recent study also revealed that the overexpression of the *CsFUS3* gene of the B3 TF family can promote somatic embryo development in citrus; this is consistent with the regulation of the *FUSCA3* gene in Arabidopsis, which has been found to regulate seed development [78,79]. Additional studies have found that *ARR12* in Arabidopsis is a central regulator of callus formation and stem regeneration; *ARR1* is a strong inhibitor of this process, counteracting the positive effects of *ARR12* under different N concentrations [80]. The findings of this study suggest that the identification of TFs is important in understanding N absorption and assimilation, which will contribute to improvements in crop NUE.

In this study, a total of 36 TFs (e.g., Nin_like, G2like, B3, HSF, Trihelix, GRAS, ARRB, MYB, WRKY, ERF, and bZIP) were found to be associated with N metabolism regulation (Figure 7E–H). Interestingly, among the identified TFs, some negatively regulate the target genes related to N metabolism, such as *AUR62007169(Trihelix)*, *AUR62032797(G2-like)*, *AUR62000210(HSF)*, and *AUR62011826(B3)*; in contrast, other TFs showed positive regulatory roles (Figure 7E,F). An analysis of the potential interactions among these TFs was conducted, and the results point to a complex regulatory network among them. For example, *AUR62032797(G2-like)* and *AUR62011826(B3)* were negatively regulated by other TFs. Some TFs were chosen for expression analysis using qRT-PCR, and the results showed that their expressions were consistent with the expressions detected via RNAseq (Appendix A). We also further analyzed the results using phylogenetics to indicate that the detected TFs in quinoa had the conserved structure of TFs. Their function was found to be associated with the regulation of N absorption and assimilation in the model plant as well as other major crops (Appendix A). The resulting phytogenic association also suggests that the detected TFs in this study have similar functions; however, the predicted TFs need further investigation through molecular biology methods.

## 4. Materials and Methods

### 4.1. Plant Growth and Experimental Design

Three Bolivian quinoa cultivars, W32, W19, and W25, with white-colored seeds, belonging to a highland ecotype, were used in Yunnan Agricultural University (YNAU), followed by 10 successive generations through selfing. W32 was grown under hydroponic conditions in the Hydroponics Laboratory of Chemistry Building, YNAU, for better performance under hydroponic cultivation conditions. W19 and W25 were directly cultivated in the field of YNAU in the autumn–winter season. Based on a Hoagland nutrient solution, the different concentrations of NH_4_^+^ (with 0, 8, and 21 mmol/L) were designed as the only nitrogen source for W32 cultivation in hydroponic conditions to easily determine the root phenotypes. In detail, hydroponic boxes (350 × 265 × 130) equipped with an electric drill for aeration were used, in which polyvinyl chloride (PVC) foam boards were placed with 120 holes made on each foam board for quinoa cultivation. The nutrient solution contained the following constituents: 1 mM KH_2_PO_4_, 2 mM MgSO_4_.7H_2_O, 4 mM CaCl_2_, 5 mM KCl, 0/4/10.5 mM (NH_4_)_2_SO_4_, 5 µM KI, 1 µM Na_2_MO_4_.2H_2_O, 100 µM H_3_BO_3_, 0.1 µM CuSO_4_.5H_2_O, 30 µM ZnSO_4_.7H_2_O, 50 µM Fe_3_ citrate, and 0.1 µM CoCl_2_.6H_2_O. The pH was adjusted to 6 ± 0.2, and the nutrient solutions were changed every 4 d. When the W32 quinoa seedlings reached the six-leaf stage with continuous cultivation in the Hoagland solution for about 21 d after germination, seedlings with a similar appearance (uniform in both roots and leaves) were selected to grow without N for 1 week. Then, they were transferred to solutions containing different NH_4_^+^ concentrations (0, 8, and 21 mmol L^−1^) for 27 d. The field-planted variants, W19 and W25, comprised a total of 6000 plants/667 m^2^ with a total nitrogen supply of 9 kg/667 m^2^ and N:P_2_O_5_:K_2_O = 2:1:2 until seed harvesting. Each treatment comprised three biological replicates with at least three plants per replicate. The roots (Rs), leaves (Ls), and panicles (Ps) of quinoa were sampled at the end of the treatments, frozen immediately in liquid nitrogen, and stored at −80 °C for further analysis.

### 4.2. Illumina RNAseq and Analysis

The total RNAs were extracted from the roots, leaves, and panicles samples of the quinoa plants through an RNAPrep pure plant kit (TianGen, Beijing, China); then, they were reverse-transcribed to cDNA using a GoScript™ Reverse Transcription Mix Kit, Random Primer (Promega, Beijing China), according to the manufacturer’s protocols. cDNA (~10 μg) was sequenced using the Illumina Genome Analyzer II platform (Illumina, San Diego, CA, USA) with libraries of 150 bp × 2 paired-end (PE) reads. All RNAseq data (Appendix A) were evaluated with FastQC (https://github.com/topics/fastqc, accessed on 11 August 2023) and then filtered with Fastp v0.23.4 (https://github.com/OpenGene/fastp, accessed on 12 August 2023) to obtain clean data with default parameters. The clean reads were mapped to the reference genome (https://phytozome-next.jgi.doe.gov/info/Cquinoa_v1_0, accessed on 14 August 2023) using hisat2 v2.2.1 [81]. Expression quantification was performed through scripts of FeatureCounts [82] to obtain a matrix of TPM (standardized expression units of transcripts per kilobase of exon model per million mapped reads). All gene expression analyses were based on the TPM matrix. The predicted protein sequences of quinoa genomes were submitted to eggNOG (http://eggnog-mapper.embl.de/, accessed on 18 August 2023) as well as the NCBI NR database (https://www.ncbi.nlm.nih.gov/ accessed on 13 July 2024) for functional annotations. All the RNA sequencing data have been uploaded to the CNCB with the following IDs: CRA017546 and CRA014457 (https://ngdc.cncb.ac.cn/?lang=zh, accessed on 11 July 2024).

### 4.3. Identification of the AMT Genes in the Genome of Chenopodium quinoa

The genome sequence and its generic feature format (GFF3) annotation file of quinoa, obtained from the genome database (https://phytozome-next.jgi.doe.gov/info/Cquinoa_v1_0, accessed on 13 July 2024), were used to identify the longest transcript and protein of coding genes. To identify the candidate *CqAMT*s, the protein sequences of six *AtAMTs* of *A. thaliana* were downloaded from TAIR10 (https://www.Arabidopsis.org/, accessed on 13 July 2024), and 10 rice *OsAMTs* were downloaded from the Phytozome (https://phytozome-next.jgi.doe.gov/, accessed on 13 July 2024). These were used to search the longest homologous proteins in the quinoa genome using BLAST v2.11.0+, with a filtering standard of similarity of >40%, a coverage of >80%, and e-values of <1e^−6^ [83]. Simultaneously, the hmmmodel (ID: PF00909), obtained from the Pfam database (https://www.ebi.ac.uk/interpro/, accessed on 13 July 2024), was used to search the longest transcripts or proteins of all possible *CqAMTs* with a filtering standard of an e-value of <1e^−3^, using the software of hmmsearch v3.3.1 (http://hmmer.org/, accessed on 13 July 2024). After the intersected gene sets were obtained using BLASTP and Hmmsearch, the putative *CqAMT* family members were identified from the quinoa genome, *CqAMT1.1a* through *CqAMT3.1b*, according to the sequence homologies and based on phylogenetic analyses. To detect the gene structure variation in the *CqAMTs*, the transcript data were mapped to the genomes to locate the exons and introns and the translational region (CDS) and un-translational region (UTR). Furthermore, the ProtParam tool (https://web.expasy.org/ protparam/, accessed on 14 July 2024) was used to estimate the basic physical and chemical properties of the proteins. The TMDs of *CqAMTs* were predicted by web tools of TMHMM 2.0 (http://www.cbs.dtu.dk/services/TMHMM/, accessed on 14 July 2024) [84].

### 4.4. Characterization of CqAMTs

#### 4.4.1. Phylogenetic Analysis of CqAMTs

The full-length amino acid sequences of *AMTs* from A. thaliana [4,16,20], rice (*Oryza sativa*) [15,52], tomato (*Solanum lycopersicum*) [85,86], tea (*Camellia sinensis*) [53], and poplar (*Populus trichocarpa*) [87], along with the *CqAMTs* identified in this study, were used for the phylogenetic analyses. Their IDs are listed in Appendix A. The multiple sequence alignment at the amino acid level was performed by MUSCLE [88]; the phylogenetic tree was constructed based on referring to the maximum likelihood method of RAxML [89]. There were 1000 bootstrap replicates, and it was beautified with the iTOL online website (https://itol.embl.de/, accessed on 16 July 2024) [90].

#### 4.4.2. Conserved Motifs, Gene Structure, and Chromosomal Locations

The diagrams of gene structure, exon–intron organization, and chromosomal location were drawn using TBtools [91]. All the conserved motifs of *CqAMT* were predicted through the motif scripts (MEME) (https://meme-suite.org/meme/tools/meme, accessed on 16 July 2024). To analyze the gene structure and chromosome location of *CqAMTs*, the basic information in the GFF3 file, which was manually corrected of RNAseq data, was introduced into the Gene Structure Display Serve 2.0 software (http://gsds.cbi.pku.edu.cn/, accessed on 16 July 2024) to visualize the output.

#### 4.4.3. Cis-Element Analysis Considering CqAMT Promoters 

The 2000 bp unflanking sequences of *CqAMT*s, starting from the initiation codon (ATG) obtained with TBtools [91], were submitted to the Plant CARE databases (https://bioinformatics.psb.ugent.be/webtools/plantcare/html/, accessed on 16 July 2024) [92] for the prediction of cis-acting elements.

#### 4.4.4. Gene Duplication and Collinearity Analysis

To analyze gene duplication events, the Multiple Collinearity Scan toolkit (MCScanX) [93] with the default parameters was used; the output was visualized with Circos software 3.0 (https://github.com/node/circos accessed on 13 July 2024).

### 4.5. Quantitative Real-Time PCR (qRT-PCR) Analysis

Total RNA was extracted from the samples described above, using the Hipure HP Plant RNA Mini Kit (Magen, Shanghai, China) according to the manufacturer’s protocols. Of the extracted total RNA, ~0.9 μg was used to synthesize the first-strand cDNA using FastKing gDNA Dispelling RT Super Mix (TIANGEN, Beijing China). qRT-PCR was performed with a SYBR qPCR Master Mix (BestEnzymes Biotech Co., Ltd., Lianyungang China) in an ABI7500 StepOnePlus Real-Time PCR System (ABI, Natick, MA, USA). The PCR mixtures were first incubated in a 96-well optical plate at 95 °C for 60 s, followed by 40 cycles at 95 °C for 10 s and 60 °C for 30 s. A melt curve was drawn by the instrument default acquisition program. The expression levels were calculated using the 2^−∆∆CT^ method [94] with the primers listed in Appendix A.

### 4.6. Subcellular Localization Analysis and Prediction of CqAMT1.2a

The pCA1301-GFP vector was used as the starting vector; the coding region of *CqAMT1.2a* (excluding the stop codon) was amplified through a PCR from cDNA. This was cloned into the pCA1301-GFP vector with double digestion of both *BgI* II and *Spe* I. The resulting pCA1301-CqAMT1.2a-GFP vector (Appendix A) was transformed into the *Agrobacterium tumefaciens* strain GV3101, selected using a kanamycin-resistant form, and the results reconfirmed the transformation via PCR amplification. The re-culture of pCA1301-CqAMT1.2a-GFP/GV3101, which was harvested by centrifugation at 4000 rpm for 5 min, was suspended with a 10 mM MgC1_2_ (including 120 µM AS) solution. Using an adjusted concentration with OD600 = 0.4, the cultivated pCA1301-CqAMT1.2a-GFP/GV3101 was used to inject the lower parts of the epidermis of tobacco leaves. The injected leaves were cultured in low-light conditions for 2 d; then, the injected tobacco leaves were taken to cut into the slides. Finally, the subcellular locations of *CqAMT1.2a* were detected using GFP fluoresce under a confocal laser microscope (ZEISS, Oberkochen, Germany) and photographed to record the data. PSORT and BUSCA were used to predict the subcellular localization of *CqAMT*-encoded proteins [95,96].

### 4.7. Cloning and Functional Complementation of CqAMTs in Yeast Triple mep1/2/3 Mutant

The open reading frames (ORFs) of 11 *CqAMTs* were successfully amplified using qRT-PCR with specific primers containing the *Hind III* and *Eco RI* sites (listed in Appendix A). The gel-purified PCR products were cloned into the yeast expression vector pYES2 using a Clon Express II One Step Cloning Kit (TOLBIO, Irvine, CA, USA). The pYES2-CqAMTs plasmids were transformed into the triple mep1/2/3 deletion yeast mutant 31019b (*mep1Δ*, *mep2Δ: LEU2*, and *mep3Δ: KanMX2 and Δura3*). A mutated yeast strain will not grow normally if the concentration of NH_4_^+^ (as the sole nitrogen source) is lower than 5 mmol L^−1^ [38]. All transformants were precultured on a liquid medium supplemented with 2 mmol L^−1^ arginine (Arg) as the only N source until the OD600 value reached 0.6~0.8. The transformed yeast was concentrated to OD600 = 1.0 through centrifugation (suspended in sterile water). The cultivated yeast samples were then divided into four groups based on dilution (i.e., no dilution, as well as 10-fold, 100-fold, and 1000-fold dilution). Briefly, 5 μL aliquots of different dilutions were applied onto a solid SD medium supplemented with 2% *w*/*w* galactose and 0.3, 3, or 30 mmol L^−1^ NH_4_^+^ (NH_4_Cl applied as the sole N source) and cultured at 30 °C for 2~3 d, respectively.

### 4.8. Enzymatic Assays and Phenotypic Measurements of Quinoa Roots

The root system was scanned with a root scanner (Shanghai Zhongjing Technology Co., Ltd., Shanghai, China, ScanMaker i800 Plus), and the collected data were analyzed with WinRHIZO 2019b (Regent Instruments, Québec City, QB, Canada). The activities of glutamine synthetase (GS), glutamate synthase (GOGAT), nitrite reductase (NiR), and nitrate reductase (NR) in roots were determined using micro determination kits (Geruisi Biotechnology Co., Ltd., Suzhou, China), following the manufacturer’s protocols.

### 4.9. Prediction of Gene Regulatory Networks for Nitrogen Transportation and Assimilation

The co-expression gene module analysis was conducted with gene co-expression networks using WGCNA (v1.47) in R (Langfelder and Horvath 2008) with the following parameters: power = 9; min module size = 30; merge cut height = 0.15. The identified gene modules containing the *CqAMTs* were further used to construct the gene regulatory networks associated with nitrogen transportation and assimilation based on TFs. Their binding sites (TFBSs) were predicted in the 2000 bp upstream promoters, flanking the sequence from the start coding (ATG) of the target gene. The correlations between the expression of TF(s) and target gene(s) were investigated using the Pearson correlation analysis, conducted with the R package of psych (https://github.com/neuropsychology/psycho.R accessed on 28 July 2024), and the results were filtered with an absolute R > 0.6 and *p*-values < 0.05. Additionally, the correlations between the expressions of the putative TFs and at least three phenotypes of quinoa (i.e., fresh weight, dry weight, specific length, surface area, average diameter, and number of root tips), as well as at least three activities of GS, GOGAT, NiR, or NR in roots, were analyzed. The resulting correlation networks were visualized using Cytoscape (https://cytoscape.org/, accessed on 28 July 2024) [97].

## 5. Conclusions

In this study, the *AMT* gene family from the quinoa genome was identified. Eleven of them, with the exception of *CqMT2.2b*, had NH_4_^+^ transportation activities. Moreover, the *CqNLPs*, *CqG2Ls*, *B3TFs*, *CqbHLHs*, *CqZFs*, *CqMYBs*, *CqNF-YA/YB/YC*, *CqNACs,* and *CqWRKY* genes were predicted to be involved in the regulation of ammonium transportation and assimilation, especially in the roots. The gene regulatory networks, including TFs that target the genes associated with nitrogen transport and assimilation, as well as TFs that target TF genes in quinoa, were investigated. The findings provide valuable insights for improving NUE in quinoa as well as other crops.

## Figures and Tables

**Figure 1 plants-13-03524-f001:**
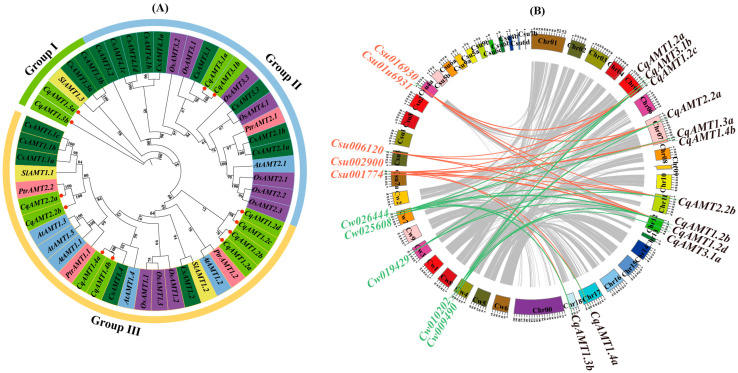
Phylogenetic tree of *AMTs* and their colinear relationship among *C. watsonii*, *C. suecicum*, and *C. quinoa*: (**A**) Phylogenetic tree of the AMT proteins of six plant species (*A. thaliana*, *O. sativa*, *S. lycopersicum*, *P. richocarpa*, *C. sinensis* var. sinensis, and *C. quinoa*). Each group is represented by a different color, and the CqAMT proteins are marked with red dots. (**B**) Reservation and loss of the *AMT* genes among *C. watsonii* (*Cw*), *C. suecicum* (*Cs*), and *C. quinoa* (*Cq*).

**Figure 2 plants-13-03524-f002:**
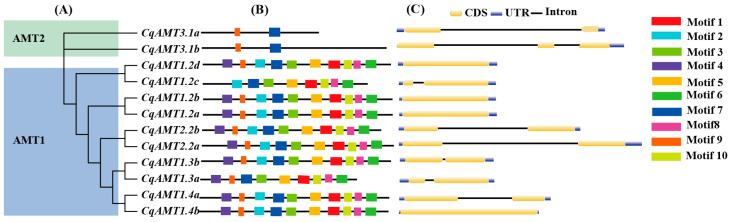
Phylogenetic relationships, conserved motifs, and gene structure analysis of *CqAMT* genes: (**A**) Phylogenetic tree of the 12 CqAMT proteins. (**B**) The conserved protein motifs were identified using MEME; each color represents a motif. The lengths of the motifs are proportional. (**C**) The exon–intron distribution of *CqAMTs* with black lines indicated introns, while exons are indicated with yellow boxes (CDS) and blue boxes (UTR).

**Figure 3 plants-13-03524-f003:**
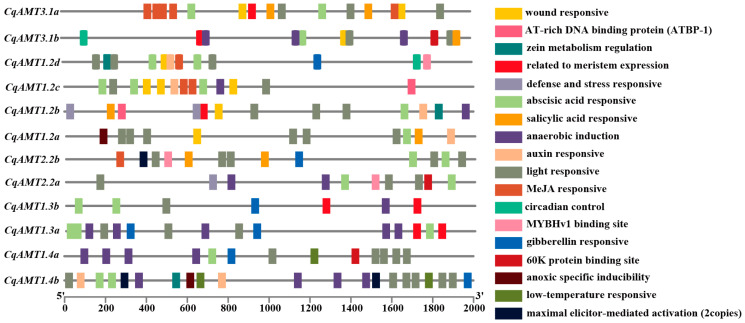
Predicted cis-elements in 12 *CqAMTs* promoters, predicted using PlantCARE.

**Figure 4 plants-13-03524-f004:**
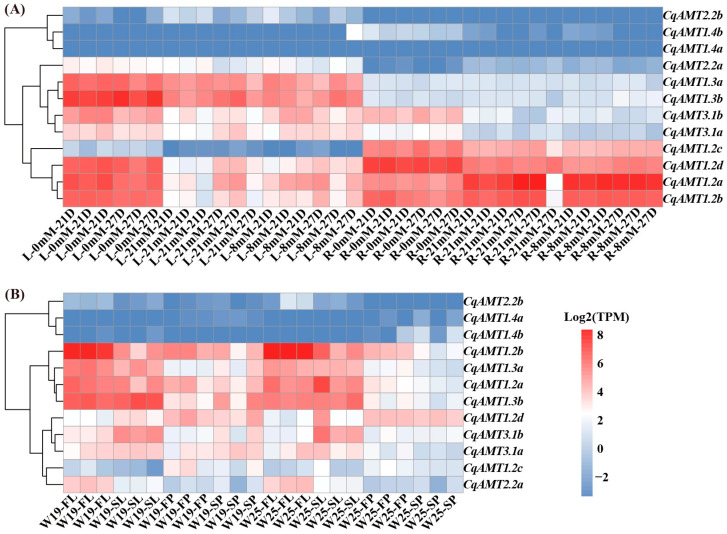
Expression patterns of *CqAMT* genes: (**A**) W32 leaves and roots under 0, 8, and 21 mM NH_4_^+^ concentrations, respectively (L-0mM-21D—leaf samples were treated with 0 mM ammonium nitrogen for 21 d, with similar descriptions as those below for L-0mM-27D, L-21mM-21D, L-21mM-27D, L-8mM-21D, and L-8mM-27D; R-0mM-21D—root samples were treated with 0 mM ammonium nitrogen for 21 d, with similar descriptions as those below for R-0mM-27D, R-21mM-21D, R-21mM-27D, R-8mM-21D, and R-8mM-27D). (**B**) *CqAMTs* expressed in different developmental reproductive stages of both W19 and W25 planted in field (W19-FL—leaves of W19 at the flower development stage; W19-SL—leaves of W19 at the seed-filling stage; W19-FP—panicles of W19 at the flowering stage; W19-SP—panicles of W19 at the seed formation stage); W25 samples were labeled similarly as W19 samples.

**Figure 5 plants-13-03524-f005:**
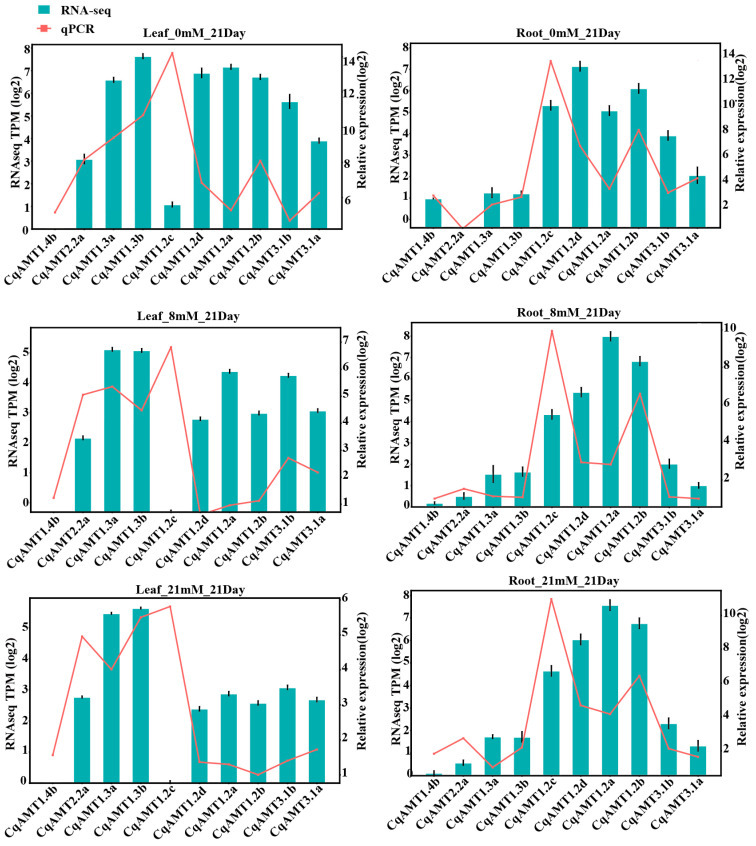
qRT-PCR analysis of the 10 *CqAMT*s in both leaves and roots under hydroponic cultivation of W32 after 21 d with 0, 8, and 21 mM NH_4_^+^ concentrations.

**Figure 6 plants-13-03524-f006:**
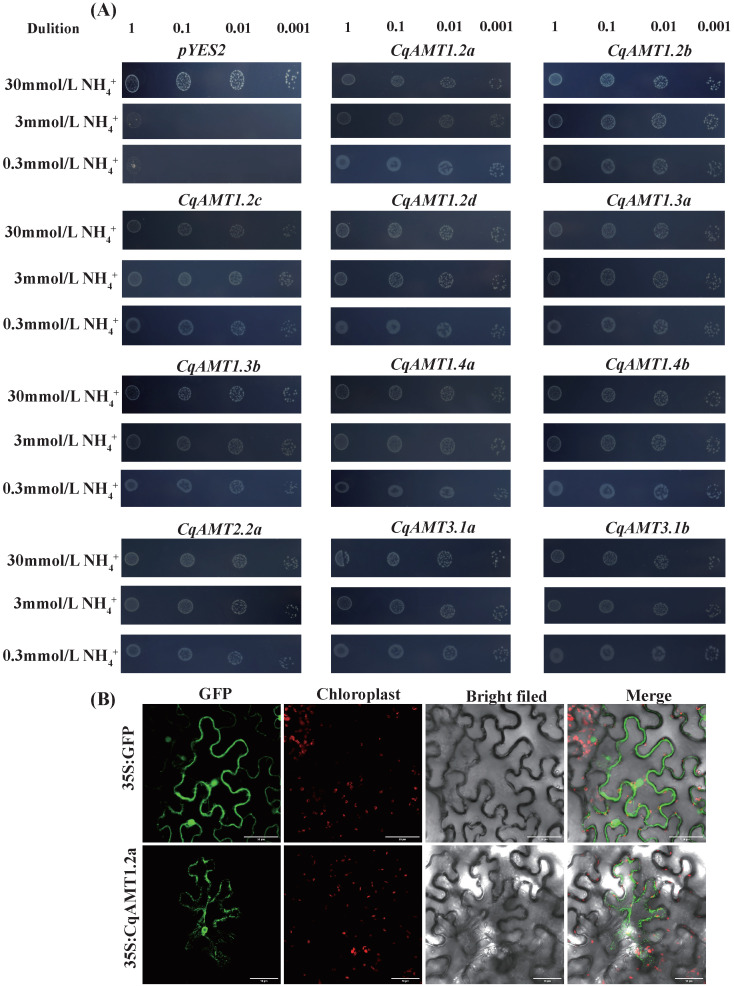
Functional verification of 11 *CqAMTs* and *CqAMT1.2a* subcellular location detection: (**A**) Growth of the yeast mutants (31019b) was complemented via heterologous expression of *CqAMTs*. The yeast mutant strain (31019b) was transformed with the empty vector pYES2, or 11 *CqAMTs* expression vectors, namely CqAMT2.2a-pYES2, CqAMT1.3a-pYES2, CqaMT1.4a-pYES2, CqAMT3.1b-pYES2, CqAMT1.2c-pYES2, CqAMT1.2a-pYES2, CqAMT1.4b-pYES2, CqAMT1.2b-pYES2, CqAMT1.2d-pYES2, CqAMT3.1a-pYES2, and CqAMT1.3b-pYES2. The mutant 31019b transformed with pYES2 was used as a negative control. The transformants were grown on the SD medium at 30 °C for 2–3 days. (**B**) Subcellular localization detection of *CqAMT1.2a* was performed by fusing the expression with *GFP* in tobacco leaves.

**Figure 7 plants-13-03524-f007:**
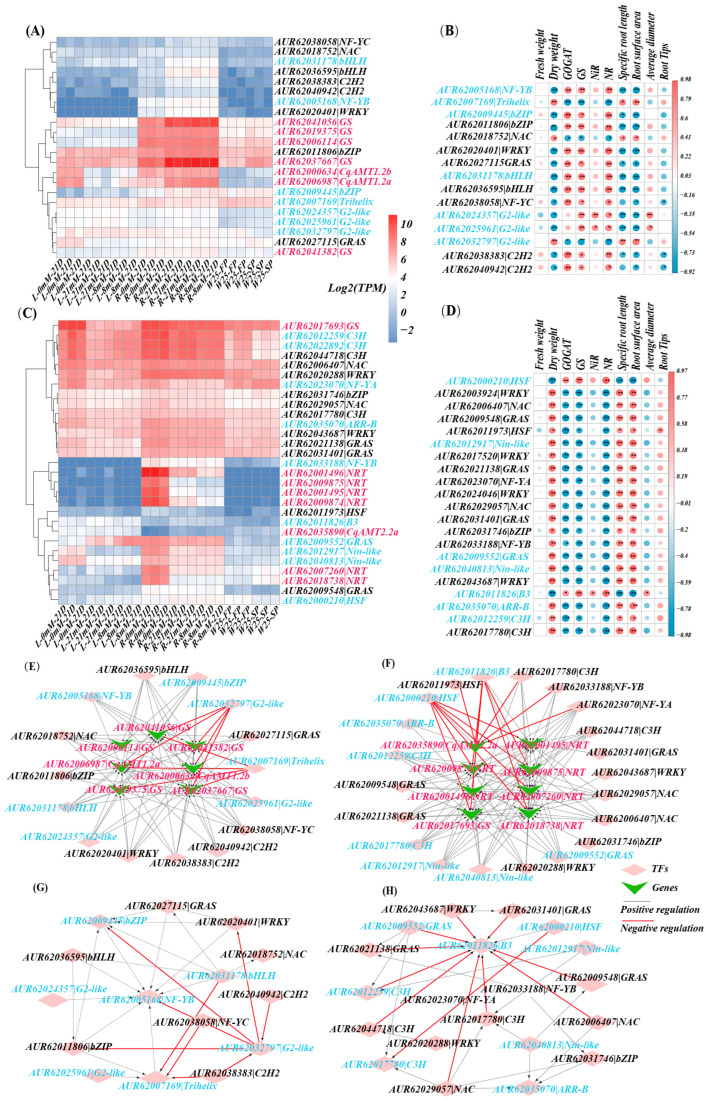
Co-expression network construction and identification of TFs: (**A**) The expression levels of screened TFs and genes related to nitrogen metabolism in different tissues and different nitrogen concentration samples in BGM. (**B**) The correlation between physiological traits and the expression level of screened TFs in the BGM. (**C**) The expression levels of screened TFs and genes related to nitrogen metabolism in different tissues and samples using different nitrogen concentrations in TGM. (**D**) The correlation between physiological traits and the expression level of screened TFs in TGM. (**E**) Co-expression network of top 15 TFs in BGM. (**F**) Co-expression network of top 21 TFs in TGM. (**G**) TF-TF co-expression network of BGM. (**H**) TF-TF co-expression network of TGM. The * symbol represents 0.01 < *p* < 0.05. The ** symbol represents *p* < 0.01.

**Figure 8 plants-13-03524-f008:**
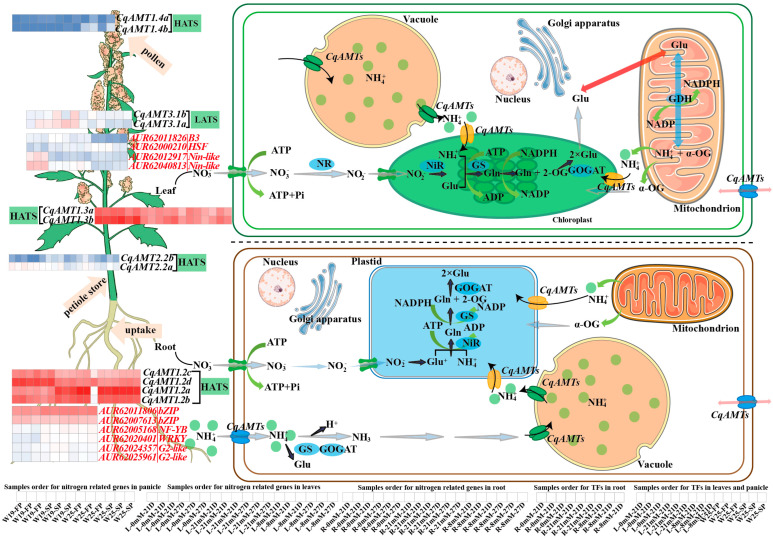
Nitrogen uptake and utilization mechanism in *Chenopodium quinoa* [39,40,41]. NO_3_^−^—nitrate; NO_2_^−^—nitrite ion; NH_4_^+^—ammonium; NRT—nitrate transporter; NiR—nitrite reductase; NR—nitrate reductase; Gln—glutamine; Glu—glutamic acid; GS—glutamine synthase; GOGAT—glutamate synthetase; GDH—glutamate dehydrogenase; α-OG—α-ketoglutaric acid; NADP—nicotinamide adenine dinucleotide phosphate.

**Table 1 plants-13-03524-t001:** Physical and chemical properties of 12 *CqAMTs* in quinoa.

Gene Name	Gene ID	Chr	CDS (bp)	AA	MW (kDa)	pI	TMD	Predicted Subcellular Localization
*CqAMT1.2a*	AUR62006987	5	1500	499	53.38	7.1	11	Plasma membrane
*CqAMT1.2b*	AUR62000634	12	1497	498	53.31	7.1	11	Plasma membrane
*CqAMT1.2c*	AUR62006988	5	1485	494	46.52	7.03	10	Plasma membrane
*CqAMT1.2d*	AUR62000635	12	1488	495	53.06	7.03	11	Plasma membrane
*CqAMT1.3a*	AUR62001658	7	1389	462	44.02	5.47	7	Plasma membrane
*CqAMT1.3b*	AUR62020119	18	1389	462	47.84	5.47	9	Plasma membrane
*CqAMT1.4a*	AUR62008822	17	1500	499	53.47	6.04	11	Plasma membrane
*CqAMT1.4b*	AUR62025419	7	1500	499	53.45	5.78	11	Plasma membrane
*CqAMT2.2a*	AUR62035890	7	1518	505	54.12	6.3	9	Plasma membrane
*CqAMT2.2b*	AUR62035356	11	1302	433	50.54	6.58	9	Plasma membrane
*CqAMT3.1a*	AUR62039048	12	933	310	33.79	9.52	7	Plasma membrane
*CqAMT3.1b*	AUR62004789	5	1461	486	52.71	8.87	11	Plasma membrane

## Data Availability

The raw sequence data reported in this paper have been deposited in the Genome Sequence Archive in the National Genomics Data Center, China National Center for Bioinformation/Beijing Institute of Genomics, Chinese Academy of Sciences (GSA: CRA017546 and CRA014457) that are publicly accessible at https://ngdc.cncb.ac.cn/gsub/ (accessed on 28 July 2024).

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
