# Peer review of "The Identification of AMT Family Genes and Their Expression, Function, and Regulation in Chenopodium quinoa"

_plants, 2024, doi:10.3390/plants13243524_

Round 1

Reviewer 1 Report

Comments and Suggestions for Authors

This article systematically introduces the information of the AMT family members in quinoa and preliminarily elucidates the biological functions of these family members in the plant's absorption of ammonium (NH4+). However, there are still the following issues that need to be revised:

1、The summary of the main work in the title is missing; it should explicitly state the biological functions rather than simply describing what work was done. The title should reflect the content of the research conclusions.

  1. Table 1 is poorly formatted; and realign the relevant content.
  2. In Figure 6, AMT1.2b should be consistent with other names: CqAMT1.2b. It is very unfortunate that the cloning of 2.2b failed, and the subcellular localization did not show all family members. Suggestion: Add the predicted subcellular localization of all members in Table 1.
  3. Figure 8 would be more appropriate in the discussion section, where some data results can be displayed. The right side of Figure 8 should have references.
  4. The overall size of the yeast experiment images in Figure 6 is too large; please reformat.
  5. In Figure 5, there is a significant difference between the RNA seq and qRT-PCR for most family members. The RNA seq can be displayed in a bar graph, and the qRT-PCR can be displayed in a line graph within a subplot. The label qPCR in the figure should be written in full. Additionally, data for 1.4a and 2.2b are missing.

Reviewer 2 Report

Comments and Suggestions for Authors

The manuscript is well-written and provides a comprehensive investigation into the CqAMT gene family in quinoa, exploring its phylogenetic relationships, structural features, expression patterns, and potential roles in nitrogen transport and assimilation. The integration of RNA-seq data, cis-element analysis, and yeast functional assays adds depth to the study, offering valuable insights into the molecular mechanisms underlying ammonium transport in quinoa. The results contribute to our understanding of nitrogen use efficiency (NUE) in quinoa and lay the groundwork for future breeding strategies.

However, a few aspects could benefit from further elaboration or clarification to enhance the manuscript’s impact and readability:

-      The layout of Table 1 is inconsistent, with varying cell widths and poor alignment. Please adjust the table to ensure uniform column widths and improve the overall formatting for better readability and presentation.

-      The resolution of most figures appears to be somewhat low. Specifically, Figure 1(B) makes it difficult to clearly identify the chromosomes. Please consider improving the resolution and clarity of the figures to enhance their readability and visual quality.

-      The figure legends throughout the manuscript are insufficient and do not provide enough detail to explain the figures independently. A figure legend should be comprehensive enough to stand alone. In particular, Figure 4 appears to have discrepancies between the RNA-seq sample names and those described in the figure legend. This inconsistency needs to be addressed. Overall, improvements are required to ensure that the figure legends are self-explanatory and align with the data presented in the figures.

-      The discussion provides detailed insights into the expression patterns of CqAMT genes, the phylogenetic relationships among AMT genes, cis-element analysis, and the roles of related transcription factors (TFs). However, it does not clearly expand on the functional implications of CqAMT genes based on yeast experimental results, nor does it adequately consider the differences between plant and yeast systems. Are there no systematic differences between plants and yeast that should be considered in the interpretation? Does yeast have a GS/GOGAT pathway? If not, is there sufficient evidence to explain NH4 assimilation mediated by CqAMTs in the yeast model?
